# Function and Characterization Analysis of BodoOBP8 from *Bradysia odoriphaga* (Diptera: Sciaridae) in the Recognition of Plant Volatiles and Sex Pheromones

**DOI:** 10.3390/insects12100879

**Published:** 2021-09-28

**Authors:** Yuting Yang, Liang Luo, Lixia Tian, Changwei Zhao, Hongli Niu, Yifeng Hu, Caihua Shi, Wen Xie, Youjun Zhang

**Affiliations:** 1Forewarning and Management of Agricultural and Forestry Pests, Hubei Engineering Technology Center, Yangtze University, Jingzhou 434025, China; yangyuting198@163.com (Y.Y.); Luoliang112@163.com (L.L.); Zhao97640094@163.com (C.Z.); NHL213265@163.com (H.N.); huyi19913327@163.com (Y.H.); Shicaihua1127@163.com (C.S.); 2Institute of Plant and Environment Protection Beijing Academy of Agriculture and Forestry Sciences, Beijing 100081, China; Tianlixia0829@163.com; 3Department of Plant Protection, Institute of Vegetables and Flowers, Chinese Academy of Agricultural Sciences, Beijing 100081, China

**Keywords:** *Bradysia odoriphaga*, odorant-binding protein, competitive binding assay, homology modeling, molecular docking

## Abstract

**Simple Summary:**

*Bradysia odoriphaga* (Diptera: Sciaridae) is an important underground pest in Chinese chives. Chemical pesticides are the main prevention and control method, however, but this method not only leads to the increase of the insect’s resistance, but also causes pesticide residues and pollutes the environment. Previous studies have shown that olfaction plays a crucial role in the recognition of plant volatiles and sex pheromones, but the mechanism of olfactory action is still unclear. In the present study, Real-time PCR (qRT-PCR) analysis revealed that BodoOBP8 was highly expressed in the antennae of both sexes, and speculated that it is very likely to participate in the olfactory process. Then we used prokaryotic expression, fluorescence competitive binding, homology modeling, and molecular docking to prove its olfactory function. The results of this study increase our understanding of the binding of BodoOBP8 with plant volatiles and sex pheromone, facilitating the development of novel ways to control *B. odoriphaga*.

**Abstract:**

The belowground pest *Bradysia odoriphaga* (Diptera: Sciaridae) has a sophisticated and sensitive olfactory system to detect semiochemical signals from the surrounding environment. In particular, odorant-binding proteins (OBPs) are crucial in capturing and transporting these semiochemical signals across the sensilla lymph to the corresponding odorant receptors. In this study, we cloned a full-length cDNA sequence of *BodoOBP8* from *B. odoriphaga*. Real-time PCR (qRT-PCR) analysis revealed that *BodoOBP8* has the highest expression levels in males, with more pronounced expression in the male antennae than in other tissues. In this study, the recombinant protein BodoOBP8 was successfully expressed by a bacterial system to explore its function. Competitive binding assays with 33 host plant volatiles and one putative sex pheromone (n-heptadecane) revealed that purified BodoOBP8 strongly bound to two sulfur compounds (methyl allyl disulfide and diallyl disulfide) and to n-heptadecane; the corresponding dissolution constants (Ki) were 4.04, 6.73, and 4.04 μM, respectively. Molecular docking indicated that Ile96, Ile103, Ala107, and Leu111, located in the hydrophobic cavity of BodoOBP8, are the key residues mediating the interaction of BodoOBP8 with two sulfur compounds (methyl allyl disulfide and diallyl disulfide) and n-heptadecane. These results show that BodoOBP8 plays a role in the recognition of plant volatiles and sex pheromones, suggesting its application as a molecular target for the screening of *B. odoriphaga* attractants and repellents and facilitating a new mechanism of *B. odoriphaga* control.

## 1. Introduction

The belowground pest species *Bradysia odoriphaga* Yang et Zhang (Diptera: Sciaridae) mainly feeds on Liliaceae crops such as Chinese chives, garlic, and ginger, but it also damages lettuce and radish [1]. The larvae feed on the bulb parts of the host plant, causing quantitative and qualitative losses and, eventually, plant death [2]. The larvae live in the soil, and the species has a short reproduction cycle, impeding efficient control strategies [1]. Currently, chemical insecticides are mainly used to control *B. odoriphaga*. However, the long-term and extensive use of such compounds will not only lead to an increased resistance of *B. odoriphaga* larvae, but also to pesticide residues in host plants, with negative consequences for the ecological environment [3]. In this sense, new prevention and control strategies for *B. odoriphaga* need to be found.

More recently, a series of studies showed that insect olfaction could distinguish various hosts based on plant volatiles and insect pheromones [4,5,6,7,8,9], and host plant volatiles and insect pheromones to control insect pests have been widely used. For example, the host plant volatiles undecan-2-one, nonan-2-one, and 2-nonyl acetate have been applied as repellents for *Aedes aegypti* [10], and (Z)-3-hexenol, (Z)-3-hexenyl acetate, (Z)-3-hexenyl hexanoate, decanal, and tetradecanol have been used in detecting *Sitona lineatus* [11], *Agrilus planipennis* [12], and *Agrilus mali* [13], respectively. The compounds 2-nonanone and 2-undecanone could stimulate a positive oviposition response to *Ceratitis capitata* [14]. Additionally, the insect pheromones 4-methyl-3,5-heptanedione and (E8, E10)-dodecadienol have been used as attractants for *Sitona lineatus* [11] and *Cydia pomonella* [15]. In our previous study, we have shown that the volatiles methyl allyl disulfide and diallyl disulfide, emitted from Chinese chives, could regulate host location and oviposition behavior in *B. odoriphaga* adults [16]. In another study, the sex pheromone n-heptadecane could regulate the behavior of *B. odoriphaga* male adults [17]. The activity of *B. odoriphaga* larvae is low, and the larvae rely on adults to select suitable host plants, mates, and oviposition sites and to provide the necessary conditions for the survival and reproduction of the offspring [1]. In this sense, the development of a new control strategy for *B. odoriphaga* adults, based on host plant volatiles and sex pheromones, is a promising approach.

Insect olfaction systems are crucial in insect olfaction recognition, especially in host-seeking, mating, selecting oviposition sites, and avoiding predators [14,18,19]. The involved odorant-binding proteins (OBPs) and chemosensory proteins (CSPs) are the two most important proteins in the olfaction system; they bind to and transport the odor molecules from outside and pass through the lymph to the odorant receptors (ORs), where they perform a series of physiological processes [14,20,21,22]. Insect OBPs are a water-soluble protein family and consist of 150–220 amino acid residues and six highly conserved cysteines [23,24,25]. Analysis of insect OBPs has shown that they contain six α-helices, which can be partitioned into three pairs of interlocked disulfide bridges which are further folded to form a cavity that can bind to hydrophobic ligands [26,27]. Analysis of the expression patterns of the OBP genes in different insect species can provide clues to clarify the functions of these genes [28,29]. Previous studies have indicated that OBPs are widely expressed in the olfactory organs of insects, indicating that they participate in insect olfactory processes. For example, the OBP18 of *Helicoverpa assulta* is mainly expressed in the antennae and involved in olfactory function [30]. The OBP3 of *Bemisia tabaci* is mainly expressed in the head and plays a role in recognizing host plant odor [31], whereas the OBP1 of *Monochamus alternatus* is highly expressed in the antennae and involved in host-seeking [32]. In addition, insect OBPs could also have other functions, when they are in body and not in antennae, suggesting a possible new function not related to chemoreception. For example, OBP1 and OBP3 of *Acyrthosiphon pisum* [33]; OBP3 and OBP8 of *Megoura viciae* [34]. To date, however, few studies have investigated the olfactory gene functions of *B. odoriphaga*, and only the functions of BodoOBP1, BodoOBP2 [35], and BodoOBP5 [36] have been clarified so far.

In our previous study, BodoOBP8 was extensively expressed in the antennae, irrespective of the sex (Figure 1), which is in agreement with the findings for the *B. odoriphaga* antennae transcriptome [37]; we, therefore, suspected that BodoOBP8 participates in the olfaction function of *B. odoriphaga*. To identify the BodoOBP8 involved in the olfaction function of *B. odoriphaga*, we expressed BodoOBP8 in vitro and performed fluorescence binding assays to determine its binding affinities for 33 plant volatiles and one sex pheromone. Via homology modeling and molecular docking, we predicted that BodoOBP8 is responsible for the key amino acids of the binding candidate ligands. In this sense, determining the BodoOBP8 functions could facilitate the development of ecologically safe control strategies for this widespread belowground pest.

## 2. Materials and Methods

### 2.1. Insect Rearing and Chemical Ligands

Larvae of *B. odoriphaga* were obtained from the Shunyi Farm in Beijing City, China, and reared with Chinese chive in a plastic chamber under a 16:8 (L:D) h photoperiod at 25 ± 1 °C and 75 ± 5% humidity [2]. Overall, 10 *B. odoriphaga* were collected for each developmental stage (eggs, larvae, pupae, and adults) for RNA isolation, and after emergence, 1000 *B. odoriphaga* were collected for dissecting various tissues, namely antennae, heads (without antennae), abdomen, and carcass (legs + wings + thorax) for RNA isolation. For the competitive binding assays, all tested compound ligands were obtained from Sigma-Aldrich (St. Louis, MO, USA) (Appendix A).

### 2.2. Cloning, Sequencing, and Phylogenetic Analysis of BodoOBP8

The cDNA sequence of BodoOBP8 (accession number: MG544128.1) was obtained from the *B. odoriphaga* antennae transcriptome [23].We predicted the open reading frames (ORFs) and N-terminal signal peptides of the sequences using the ORF Finder (https://www.ncbi.nlm.nih.gov/orffinder, accessed date: 10 December 2020) and the SignalP 5.1 server (http://www.cbs.dtu.dk/services/SignalP/, accessed date: 11 December 2020). In addition, the molecular weights and theoretical isoelectric points of proteins were calculated online using the ExPASy tool “ProtParam” (https://web.expasy.org/protparam, accessed date: 11 December 2020). The phylogenetic tree was generated in MEGA 6.0. The protein names and sequences of the 228 OBPs used in this analysis are listed in Appendix A.

### 2.3. qRT-PCR

The transcript levels of BodoOBP8 in different developmental stages and different tissues of males and females were assessed via qRT-PCR (Applied Biosystems, Waltham, CA, USA). The qRT-PCR reactions were performed as described in our previous study [36]. The primers used for qRT-PCR are shown in Table 1; RPL18 and RPS15, EF1 and ACT were used as reference genes for different stages and different tissues, respectively. The data were analyzed using the threshold cycle number (CT) and the 2^−ΔΔ^ Ct method [38]. All real-time qPCR assays were performed using three biological replicates, and significant differences in the expression patterns of the BodoOBP8 gene in different stages and tissues were analyzed using one-way analysis of variance (ANOVA), followed by the LSD test (least significant difference test); significance was determined as *p* < 0.05. All data were analyzed using the SPSS 20.0 software (SPSS Inc., Chicago, IL, USA).

### 2.4. Bacterial Expression and Purification of BodoOBP8

The primers for BodoOBP8 were designed using Primer 5.0 (Table 2). As described in Yang et al. (2021) [31], the PCR products of BodoOBP8 were linked to the bacterial expression vector pBM30 (Biomed, Beijing, China) for expression in prokaryotic BL21 (DE3) cells. The positive colony was incubated overnight in LB medium containing ampicillin (100 mg/mL) in a shaker at 220 rpm and 37 °C. Subsequently, the cultures were diluted to 1:100 with fresh LB liquid medium (500 mL) until an OD_600_ value of 0.6–0.8 was obtained. To stimulate the expression of recombinant protein, 1 mM isopropyl β-d-1-thiogalactopyranoside (IPTG) was added to the cultures, followed by cultivation for 6 h at 37 °C. To obtain bacterial cells, the mixture was centrifuged at 12,000 rpm for 30 min, and the cells were resuspended in phosphate-buffered saline (PBS) buffer. After this, the obtained cells were sonicated on ice for 30 min, and Ni ion affinity chromatography (GE Healthcare, Chicago, IL, USA) was applied to purify the collected supernatant twice after centrifugation. According to the instructions of the manufacturer, the His tag of the recombinant protein was removed via recombinant enterokinase (Novagen, Bloemfontein, South Africa). The quality and concentration of the recombinant protein were checked by SDS-PAGE and the BCA Protein Assay Kit (Beyotime, Shanghai, China). The obtained recombinant protein was stored at −80 °C.

### 2.5. Fluorescence Binding Assays

The method of fluorescence competitive binding is based on the procedure described by Yang et al. (2021) [31]. For the binding assays, all test ligands (Appendix A) were dissolved in methanol, and *N*-phenyl-1-naphthylamine (1-NPN) was used as a fluorescent probe. The dissociation constants of competitive ligands were calculated from the corresponding IC_50_ (displacement of more than 50% of 1-NPN) values, applying the following equation: Ki = [IC_50_]/1 + [1-NPN]/K_1-NPN_, where [1-NPN] is the free concentration of 1-NPN and K_1-NPN_ is the dissociation constant of the complex protein/1-NPN [39].

### 2.6. Homology Modeling and Molecular Docking

The modeling structure of BodoOBP8 was obtained with a template of CquiOBP1 (3OGN), using the Online Swiss-model software. The binding cavity was predicted by an automobile mode by the SYBYL 7.3 software. The molecular conformations of diallyl disulfide, methyl allyl disulfide, and n-heptadecane were constructed by Sketch mode and optimized using the Tripos force field and Gasteiger–Hückel charge. For molecular docking modeling, the Surflex-Dock of SYBYL 7.3 was employed. All molecular modeling between the putative BodoOBP8 protein and ligands was conducted on the Silicon Graphics^®^ (SGI) Fuel Workstation (Silicon Graphics International Corp., Milpitas, CA, USA).

## 3. Results

### 3.1. Characteristics, Sequence, and Phylogenetic Analysis of BodoOBP8

A full-length cDNA sequence BodoOBP8 was cloned using specific primers (Table 1) and verified by DNA sequencing. Based on the results, the full-length ORF consisted of 435 nucleotides and encoded 144 amino acid residues; the predicted molecular weight (MW) of BodoOBP8 was 16.56 kDa. The N-terminal of BodoOBP8 contained a signal peptide consisting of 17 amino acid residues and contained the six typical conserved cysteines of OBPs (Figure 2); the calculated isoelectric point (PI) of BodoOBP8 was 5.78 (Table 2). The phylogenetic tree of the OBPs of *B. odoriphaga* and other insect species was constructed using the neighbor-joining method (Appendix A). The BodoOBP8 was clustered with the OBPs of the dipterans *Anopheles gambiae* OBP15/16 and *Aedes aegypti* OBP36.

### 3.2. Spatio-Temporal Expression Analysis of BodoOBP8

We applied qRT-PCR to examine the expression patterns of BodoOBP8 in different tissues and at different developmental stages. Based on the results, BodoOBP8 was expressed in all stages, with the highest expression levels in adult males (Figure 1A). The BodoOBP8 expression level was highest in male antenna tissue, followed by female antenna tissue (Figure 1B).

### 3.3. Expression, Purification, and Fluorescence Competitive Binding Assays of BodoOBP8

Based on the results, the recombinant protein BodoOBP8 could be expressed in the bacterial expression system, mainly in the inclusion body. The molecular weight of BodoOBP8 was 16.56 kDa, as indicated by SDS-PAGE (Appendix A), which was consistent with the predicted result shown in Table 2.

To further confirm the specific physiological function of BodoOBP8, 33 test ligands (including one sex pheromone) were selected to determine the binding affinity of BodoOBP8 via the fluorescence competitive binding assays. Our results show that 1-*N*-phenylnaphthylamine (1-NPN) could be well bound with BodoOBP8, and the dissociation constant (K_d_) of 1-NPN was 2.21 μM (pH 7.4) (Figure 3A). Two sulfur compounds (diallyl disulfide, methyl allyl disulfide) and one sex pheromone (n-heptadecane) exhibited higher binding affinities to the protein of BodoOBP8; the Ki values of three ligands were 4.04, 6.73, and 4.04 μM, respectively (pH 7.4) (Figure 3B). However, most ligands showed weak binding affinities for BodoOBP8 (Appendix A).

### 3.4. D model of BodoOBP8 and Molecular Docking

The sequence identity was sufficiently high, with 57.5% between the target protein BodoOBP8 and the template protein CquiOBP1 (Figure 4A). Therefore, the 3D model of BodoOBP8 was generated based on the crystal structure of CquiOBP1, which showed that the structure of BodoOBP8 shows the overall fold of “classical OBPs” and is mostly helical (Figure 4B). The 3D structure of BodoOBP8 includes six α-helices, which are located between residues Lys29-Thr49 (α1), Ala50-Glu60 (α2), Leu67-Lys79 (α3), Glu90-Asp100 (α4), Arg104-Ala112 (α5), and Pro122-His140 (α6) (Figure 3A). They also possess four conserved cysteines residues, and the entire structure is further stabilized by two interlocked disulfide bonds of Cys42-Cys69 and Cys114-Cys132 (Figure 4A). All 3D structures of the target protein BodoOBP8 and the template protein CquiOBP1 aligned well (Figure 4C), indicating that the putative model of BodoOBP8 was suitable to determine the binding mode between BodoOBP8 and its ligand.

Recently, molecular modeling has become an important method to study the binding affinity between bioactive molecules and bio-macromolecules. In this study, to better understand the potential key amino acid residues in BodoOBP8 protein, the ligand–putative BodoOBP8 protein complex was studied using molecular docking; the findings are shown in Figure 5.

The binding models for diallyl disulfide, methyl allyl disulfide, and n-heptadecane, as well as the putative BodoOBP8 protein, indicated the presence of van der Waals interaction between ligand and protein (Figure 5). Diallyl disulfide, methyl allyl disulfide, and n-heptadecane were almost located on the same binding site of BodoOBP8 because of their similar 3D structure and binding conformation, particularly their identical disulfide bond groups (Figure 5A). We observed some hydrophobic residues surrounding these three ligands, including Ile96, Ile103, Ala107, Leu111, and some aromatic residues around these three ligands, including Phe110, His130, Trp133, Tyr141, and Phe142 (Figure 5B–D).

## 4. Discussion

As the main olfactory organs of insects, antennae play an important role in behavioral responses, such as searching for hosts, mates, and oviposition sites [23,40,41]. Odorant-binding proteins play an important role in the olfactory system, assisting insect antennae in recognizing and binding to external semiochemicals, which are then delivered to the ORs via the lymph [18,42,43,44,45]. Hence, determining the expression levels of insect OBP genes in different developmental stages and tissues could help us to predict the physiological functions of these genes in insects. It is generally believed that the insect OBP genes are highly expressed in the olfactory organs, especially in the antennae, implying that they are likely involved in the recognition of host plant volatiles and sex pheromones by insects [46,47,48]. For example, OBP11 in *Adelphocoris lineolatus* preferably binds to quercetin [49], OBP14 in *Holotrichia parallela* has an affinity with 6-methyl-5-heptene-2-one [50], whereas OBP1-3 in *Cylas formicarius* not only binds to the sex pheromone ((Z)-3-dodecen-1-yl (E)-2-butenoate), but also to host plant volatiles (butyl acetate, *cis*-3-hexen-1-ol, 2-hexanone, and β-ionone) [51]. In our study, interestingly, BodoOBP8 was more significantly expressed in the male antennae, suggesting that it is involved in the recognition of host plant volatiles and sex pheromones by *B. odoriphaga* adults. This result is consistent with the findings of previous studies [52,53].

To further focus on the olfactory functions of BodoOBP8 in *B. odoriphaga*, we successfully purified the recombinant protein BodoOBP8 and evaluated its binding abilities with Chinese chives (host plant) volatiles and sex pheromones, using fluorescence competition binding assays. Based on these results, the recombinant protein BodoOBP8 has high binding affinities (Ki < 10 μM) to diallyl disulfide, methyl allyl disulfide, and n-heptadecane. In our previous study, volatiles from Chinese chives could trigger EAG responses of *B. odoriphaga* antennae, and two sulfur compounds could attract the adults in a behavioral assay [16]. Previous studies showed that chemical compounds from host plants are assumed to be responsible for attracting or repelling insects, such as *Drosophila hydei* and *Acyrthosiphon pisum* [54]; *Gastrophysa viridula* and *Gastrophysa polygoni* [55]; *Ostrinia nubilalis* [56]. These results lead us to infer that host plant volatiles have great potential for use as attractants for *B. odoriphaga*, and two sulfur compounds can rapidly determine potentially bioactive semiochemicals. Based on a previous study, n-heptadecane in the abdomen extract of *B. odoriphaga* females could elicit significant behavioral responses of *B. odoriphaga* males [17]. The specific binding behaviors indicated that BodoOBP8 can recognize diallyl disulfide, methyl allyl disulfide, and n-heptadecane; these compounds may therefore be recognized and attracted by *B. odoriphaga*, suggesting that they play potential roles in the olfaction signals of *B. odoriphaga*.

Based on the results of the fluorescence competition binding assays, homology modeling and molecular docking were used to further identify the vital binding sites of BodoOBP8 with three ligands. The 3D structures showed that BodoOBP8 was the “classical OBP”, which contained a common fold of six α-helical domains and an internal cavity. This result is in agreement with previous findings [53]. Molecular docking analysis could predict cavities in ligand-binding proteins and the amino acid residues involved in ligand interaction; for example, residues of Lys74 and Pro121 in *Adelphocoris lineolatus* OBP5 [57], Tyr111 in *Holotrichia oblita* OBP1 [58], Lys123 in *Helicoverpa armigera* OBP7 [59], Leu5Ala and Met45Ala in *Apolygus lucorum* OBP22 [60], and Leu99, Leu103, Ala143, Tyr107, Phe142, and Trp144 in *B. odoriphaga* OBP5 [36]. Our results also show that some hydrophobic residues surrounding these three ligands, namely Ile96, Ile103, Ala107, and Leu111, that contribute to BodoOBP8 binding the ligands (Figure 4 and Figure 5). This suggests that Ile96, Ile103, Ala107, and Leu111 residues are the main sites facilitating the recognition of and binding to hydrophobic ligands by BodoOBP8.

Overall, our current study shows that BodoOBP8 is highly expressed in male antennae and preferentially binds to two sulfur compounds (methyl allyl disulfide and diallyl disulfide) and one sex pheromone (n-heptadecane). Both three-dimensional structural modeling and molecular docking experiments revealed that BodoOBP8 is a “classical OBP”, and some hydrophobic residues, including Ile96, Ile103, Ala107, and Leu111, are the vital function sites between Bodo OBP8 and three ligands. These results lead us to infer that OBP8 plays a vital role in *B. odoriphaga* olfactory recognition and further increase our understanding of the use of insect OBP genes as potential molecular targets for *B. odoriphaga* control.

## Figures and Tables

**Figure 1 insects-12-00879-f001:**
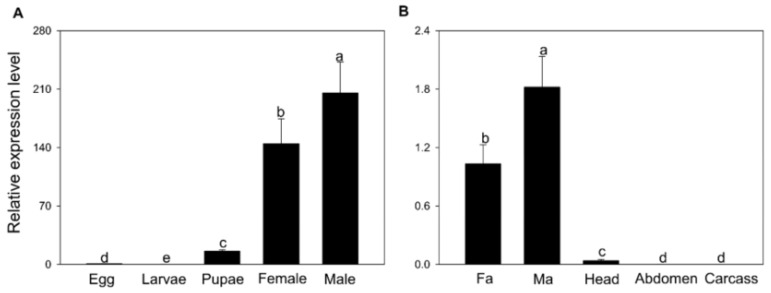
Gene expression profiling of BodoOBP8 in different developmental stages and tissues by RT-qPCR. (**A**) Eggs; larvae; pupae; female; male; (**B**) Fa: female antenna; Ma: male antenna; Head; Carcass: leg + wing + thorax; Abdomen. The expression levels were estimated using the 2^−^^ΔΔCt^ method. Values are means + SE, and means with different lowercase letters are significantly different (*p* < 0.05).

**Figure 2 insects-12-00879-f002:**
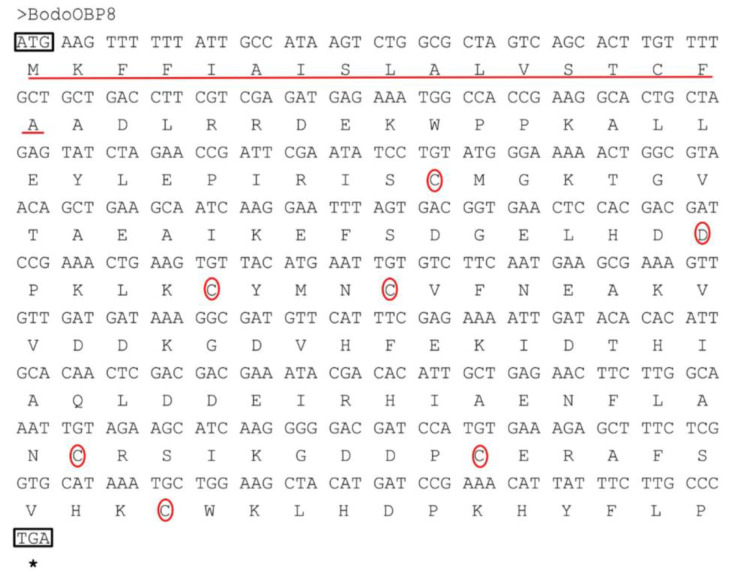
Nucleotide sequences and deduced amino acid sequences of *B. odoriphaga* OBP8. The predicted signal peptides at the N-terminus are underlined; the six conserved cysteines are marked by red boxes; the initiation and termination condons are indicated in black boxes; the temination codon is represented by an asterisk.

**Figure 3 insects-12-00879-f003:**
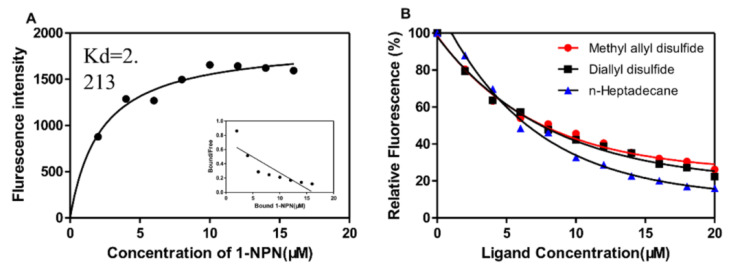
Fluorescence competitive binding assay of BodoOBP8. (**A**) Binding curves for 1-NPN and Scatchard plots; (**B**) Competitive binding curves of two sulfur compounds with BodoOBP8. (Red) methyl allyl disulfide; (black) diallyl disulfide; (blue) n-heptadecane.

**Figure 4 insects-12-00879-f004:**
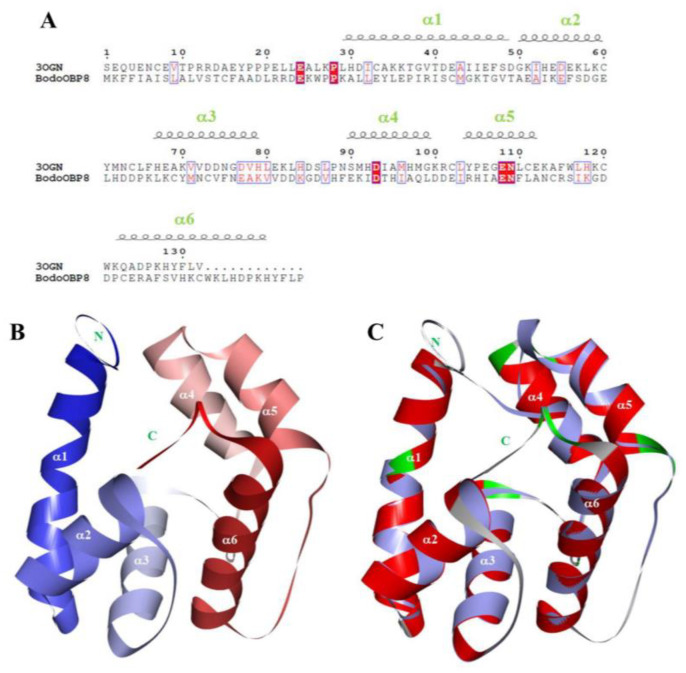
Homology modeling of BodoOBP8. (**A**) The sequence alignment between the template protein CquiOBP1 and the target protein BodoOBP8; (**B**) the 3D model of the target protein BodoOBP8 based on the crystal structure of the template protein of CquiOBP1 and the six α-helices are labeled in white; (**C**) the alignment plot of the target protein BodoOBP8 (red) and the template protein CquiOBP1 (purple, ID: 3OGN).

**Figure 5 insects-12-00879-f005:**
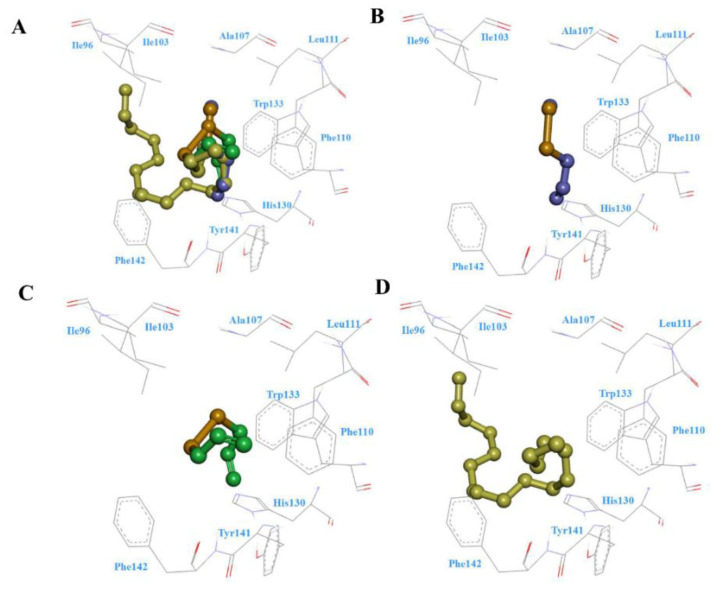
The docking mode between diallyl disulfide (purple), methyl allyl disulfide (green), and n-heptadecane (light green) with the target protein BodoOBP8. (**A**) BodoOBP8 with diallyl disulfide, methyl allyl disulfide, and n-heptadecane; (**B**) BodoOBP8 and diallyl disulfide; (**C**) BodoOBP8 and methyl allyl disulfide; (**D**) BodoOBP8 and n-heptadecane. Hydrophobic and hydrophilic residues are labeled.

**Table 1 insects-12-00879-t001:** Primers used in cloning and expression of OBP genes in *B.odoriphaga*.

Primer Name	Sequence (5′–3′)
For cloning OBP open reading frame	
BodoOBP8	GCACTTGTTCACAGTGTTTATGCTAG
	TCTGGGGAGTTTGAATTAACGAA
For spatial and temporal expression of OBP genes	
BodoOBP8	GTCGTCGAGTTGTGCAATGTG
	ACTCCACGACGATCCGAAAC
Heterologous expression of OBPs	
BodoOBP8	CACCGCTGACCTTCGTCGAGATGAGAAAT
	TCAGGGCAAGAAATAATGTTTCGGA

**Table 2 insects-12-00879-t002:** List of OBP8 genes in *B. odoriphaga*.

Gene	Acc. No	Length of ORF	Amino Acid Length	Signal Peptide	Full ORF	Isoelectric Point PI	Mw (kDa)
OBP8	MG544128.1	435	144	1–17	Yes	5.78	16.56

## Data Availability

Not applicable.

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
