# Peer review of "Function and Characterization Analysis of BodoOBP8 from Bradysia odoriphaga (Diptera: Sciaridae) in the Recognition of Plant Volatiles and Sex Pheromones"

_insects, 2021, doi:10.3390/insects12100879_

Round 1
Reviewer 1 Report
Authors performed good experiments in order to identify and characterize OBP 8 of Bradysia odoriphaga. However, some clarifications and changes are required.
SIMPLE SUMMARY
Line 22 substitute “highly expressed in the antennae both sexes, with “highly expressed in the antennae of both sexes”.
INTRODUCTION
Line 89 write “For example”.
Line 65 this sentence needs a reference “The compounds 2-nonanone and 2-undecanone could stimulate a positive oviposition response to Ceratitis capitata.
Line 72 this sentence needs a reference “The activity of B. odoriphaga larvae is low, and the larvae rely on adults to select suitable host plants, mates, and oviposition sites and to provide the necessary conditions for the survival and reproduction of the offspring.”
Line 88 authors should briefly mention that OBPs could have also other functions, when they are in body and not in antennae (as reported for example in https://doi.org/10.3389/fphys.2018.00777, doi:10.1111/1744-7917.12118)
MATERIAL AND METHODS
This is the second section, change the number of “1. Materials and Methods” and “1.1. Insect rearing and chemical ligands”
Line 118 The authors should explain why they chose specifically BodoOBP8 for the analysis.
Line 121 the authors should repeat the analysis with SignalP5.0 as it is the last version of the software.
RESULTS
This is the third section, change the number of “1. Results”.
Line 214, this sentence is not appropriate for the result section, please delete it “Recently, molecular modeling has become an important method to study the binding affinity between bioactive molecules and bio-macromolecules”.
TABLE AND FIGURES
Both tables and figures must be numerically ordered as they appear in the text. The authors wrote Table 2 and Table 1. Please be careful.
In table 2 caption, the authors should indicate what the red letters are.
In figure 2 caption, the authors inverted the samples in A with the samples in B; they used abbreviations that are not reported in the figure and letters should get in a better position on each bar.
The authors explain that the analysis was performed through the delta delta CT method, but, in order to use these equations, the efficiencies of the amplicons must be approximately equal and between the values of 0.8 and 1; moreover, they should report this information also in the material and method section.
In figure 5 caption, the explanation for A, B, C and D is missing.
DISCUSSION
Line 297 “these compounds may therefore be avoided or escaped by B. odoriphaga, suggesting that they play potential roles in the vigilance and defense of B. odoriphaga.” This sentence is not clear, moreover I don’t understand what the authors want to explain. What should the strong interaction between OBP and these compounds indicate?
Author Response
Reviewer: 1
SIMPLE SUMMARY
Line 22 substitute “highly expressed in the antennae both sexes, with “highly expressed in the antennae of both sexes”.
Corrected, please see the line 23.
INTRODUCTION
Line 89 write “For example”.
Corrected, please see the line 90.
Line 65 this sentence needs a reference “The compounds 2-nonanone and 2-undecanone could stimulate a positive oviposition response to Ceratitis capitata.
Corrected, we have added the reference, please see the lines 379-380.
Line 72 this sentence needs a reference “The activity of B. odoriphaga larvae is low, and the larvae rely on adults to select suitable host plants, mates, and oviposition sites and to provide the necessary conditions for the survival and reproduction of the offspring.”
Corrected, we have added the reference, please see the lines 350-351.
Line 88 authors should briefly mention that OBPs could have also other functions, when they are in body and not in antennae (as reported for example in https://doi.org/10.3389/fphys.2018.00777, doi:10.1111/1744-7917.12118)
Corrected, we have added the references, please see the lines 94-96, 426-430.
MATERIAL AND METHODS
This is the second section, change the number of “1. Materials and Methods” and “1.1. Insect rearing and chemical ligands”
Corrected, we have changed the number.
Line 118 The authors should explain why they chose specifically BodoOBP8 for the analysis.
Based on our previous study, in the different tissues, the expression level of BodoOBP8 was highly expressed in the antennae both sexes. Antenna is an important olfactory organ of insects. BodoOBP8 was highly expressed in Bradysia odoriphaga antennae indicated that it may be involved in the olfactory process of insects Bradysia odoriphaga. Therefore, we selected the BodoOBP8 as the research object. And the reason has been explained in our introduction, please see the lines 100-103.
Line 121 the authors should repeat the analysis with SignalP5.0 as it is the last version of the software.
Corrected, we have revised this version, please the line 124.
RESULTS
This is the third section, change the number of “1. Results”.
Corrected, we have changed the number.
Line 214, this sentence is not appropriate for the result section, please delete it “Recently, molecular modeling has become an important method to study the binding affinity between bioactive molecules and bio-macromolecules”.
Corrected, we have deleted this sentence.
TABLE AND FIGURES
Both tables and figures must be numerically ordered as they appear in the text. The authors wrote Table 2 and Table 1. Please be careful.
Corrected, we have revised the order.
In table 2 caption, the authors should indicate what the red letters are.
Corrected, we have revised the letters.
In figure 2 caption, the authors inverted the samples in A with the samples in B; they used abbreviations that are not reported in the figure and letters should get in a better position on each bar.
Corrected, we have rewritten the caption of figure 2 based on the results.
The authors explain that the analysis was performed through the delta delta CT method, but, in order to use these equations, the efficiencies of the amplicons must be approximately equal and between the values of 0.8 and 1; moreover, they should report this information also in the material and method section.
Corrected, we have rewritten the material and method section, please see the lines 133-134.
In figure 5 caption, the explanation for A, B, C and D is missing.
Corrected, we have rewritten the caption of figure 5.
DISCUSSION
Line 297 “these compounds may therefore be avoided or escaped by B. odoriphaga, suggesting that they play potential roles in the vigilance and defense of B. odoriphaga.” This sentence is not clear, moreover I don’t understand what the authors want to explain. What should the strong interaction between OBP and these compounds indicate?
Corrected, we have rewritten this sentence, please the lines 306-309.
Reviewer 2 Report
Manuscript ID: 1371880
Function and characterization analysis of BodoOBP8 from 2 Bradysia odoriphaga (Diptera: Sciaridae) in the recognition of 3 plant volatiles and sex pheromones
This is a very interesting research topic. Currently, many plant protection chemicals are being phased out in EU countries. Genetically modified plants were not approved by Europe. Therefore, there is a need to find other methods of plant protection. One of them is the natural defense mechanism of plants based on volatile organic compounds. The release of volatile organic compounds by plants or the production of phytoecdysteroids may become an important element of plant protection in the future.
This is very well organized manuscript. I found this “ms” interesting and innovative. However, a few questions must be explained more precisely.
Critical review:
- Introduction is a bit poor. I would recommend to add more scientific literature to this part of manuscript. The literature on this topic is very extensive.
- Methodology and Results are well described. Especially Results are very clear and well presented.
3. Figure 1 is unreadable. Correct it, please.
4. Discussion is unacceptable in this short way. As in the case of Introduction, I would suggest adding more references. The reader should be given a chance to compare the results with other articles.
Some other papers to add:
- Sarracenia alata Microcuttings as a source of volatiles potentially responsible for insects’ respond. Molecules 2021, 26, 2406. https://doi.org/10.3390/molecules26092406.
- Beetle Orientation Responses of Gastrophysa viridula and Gastrophysa polygoni (Coleoptera: Chrysomelidae) to a Blend of Synthetic Volatile Organic Compounds. Environ. Entomol. 49(5): 1071–1076.
- Repellent activity of plants from the genus Chenopodium to Ostrinia nubilalis Plant Protect. Sci. 54(4): 265–271.
Author Response
Reviewer: 2
1.Introduction is a bit poor. I would recommend to add more scientific literature to this part of manuscript. The literature on this topic is very extensive.
Corrected, we have added some references.
2.Methodology and Results are well described. Especially Results are very clear and well presented.
Thanks for your review.
3.Figure 1 is unreadable. Correct it, please.
Corrected, we have corrected the Figure 1.
- Discussion is unacceptable in this short way. As in the case of Introduction, I would suggest adding more references. The reader should be given a chance to compare the results with other articles.
Corrected, we have added some references. Please see the lines 299-302.